

# Parametric Decadal Climate Forecast Recalibration (DeFoReSt 1.0)

Alexander Pasternack[1], Jonas Bhend[2], Mark A. Liniger[2], Henning W. Rust[1], Wolfgang A. Müller[3], and Uwe Ulbrich[1]

[1]Institute of Meteorology, Freie Universität Berlin, Berlin, Germany
[2]Federal Office of Meteorology and Climatology (MeteoSwiss), Zürich, Switzerland
[3]Max-Planck-Institute for Meteorology, Hamburg, Germany

*Correspondence to:* A. Pasternack (alexander.pasternack@met.fu-berlin.de)

**Abstract.** Near-term climate predictions such as decadal climate forecasts are increasingly being used to guide adaptation measures. For near-term probabilistic predictions to be useful, systematic errors of the forecasting systems have to be corrected. While methods for the calibration of probabilistic forecasts are readily available, these have to be adapted to the specifics of decadal climate forecasts including the long time horizon of decadal climate forecasts, lead time dependent systematic

errors (drift), and the errors in the representation of long-term changes and variability. These features are compounded by small ensemble sizes to describe forecast uncertainty and a relatively short period for which typically pairs of re-forecasts and observations are available to estimate calibration parameters. We introduce the Decadal Climate Forecast Recalibration Strategy (*DeFoReSt*), a parametric approach to recalibrate decadal ensemble forecasts that takes the above specifics into account. *DeFoReSt* optimizes forecast quality as measured by the continuous ranked probability score (CRPS). Using a toy

model to generate synthetic forecast observation pairs, we demonstrate the positive effect on forecast quality in situations with pronounced and limited predictability. Finally, we apply *DeFoReSt* to decadal surface temperature forecasts from the MiKlip Prototype system and find consistent and sometimes considerable improvements in forecast quality compared with a simple calibration of the lead time dependent systematic errors.

## 1 Introduction

Decadal climate predictions aim to characterize climatic conditions over the coming years. Recent advances in model development, data assimilation and climate observing systems together with the need for up-to-date and reliable information on near-term climate for adaptation planning have lead to considerable progress in decadal climate predictions. In this context, international and national projects like the German initiative Mittelfristige Klimaprognosen (MiKlip) have developed model systems to produce a skillful decadal climate prediction (Pohlmann et al., 2013a; Marotzke et al., 2016).

Despite the progress being made in decadal climate forecasting, such forecasts still suffer from considerable systematic biases. In particular, decadal climate forecasts are affected by lead time dependent biases (drift) and exhibit long-term trends that differ from the observed changes. To correct these biases in the expected mean climate, bias correction methods tailored to the specifics of decadal climate forecasts have been developed (Kharin et al., 2012; Fučkar et al., 2014; Kruschke et al., 2015).



Given the inherent uncertainties due to imperfectly known initial conditions and model errors, weather and climate predictions are framed probabilistically (Palmer et al., 2006). Such probabilistic forecasts are often affected by biases in forecast uncertainty (ensemble spread), i.e. they are not reliable. Forecasts are reliable if the forecast probability of a specific event equals the observed occurence frequency on average (Palmer et al., 2008). Briefly said, if some event is declared with a certain

probability, say 80%, it should also occur on average 80% of all times such a forecast is issued. Probabilistic forecasts, however, are often found to be underdispersive/overconfident (Hamill and Colucci, 1997; Eckel and Walters, 1998), i.e. the ensemble spread underestimates forecast uncertainty and events with a forecast probability of 80% occur on average less often.

Statistical post processing (Gneiting and Raftery, 2005) can be used to optimize – or recalibrate – the forecast, e.g. reducing systematic errors, such as bias and conditional bias, as well as adjusting ensemble spread. The goal of recalibrating probabilistic

forecasts is to maximize sharpness without sacrificing reliability (Gneiting et al., 2003). A forecast is sharp if its distribution differs from the climatological distribution. For example, a constant climatological probability forecast is perfectly reliable, but exhibits small sharpness. Recalibration methods, have been developed for medium-range to seasonal forecasting; it is unclear to what extent lead time dependent biases (also called drift) and long-term trends of decadal climate forecasts can effectively be corrected. Here, we aim at adapting existing recalibration methods to deal with the specific problems found in decadal climate

forecasting: lead time and start time dependent biases, conditional biases and inadequate ensemble spread.

The most prominent recalibration methods proposed in the context of medium-range weather forecasting are Bayesian model averaging (BMA, Raftery et al., 2005; Sloughter et al., 2007) and non-homogeneous Gaussian regression (NGR, Gneiting et al., 2005). In seasonal forecasting, the climate conserving recalibration (CCR, Doblas-Reyes et al., 2005; Weigel et al., 2009) is often applied. BMA assigns a PDF to every individual ensemble member and generates a weighted average of these densities

where the weights represent the forecasting skill of the corresponding ensemble member. NGR extends traditional model output statistics (MOS) by allowing the predictive uncertainty to depend on the ensemble spread. CCR is closely related to NGR in that the forecast mean error and forecast spread are jointly corrected to satisfy the necessary criterion for reliability that the time mean ensemble spread equals the forecast root mean square error.

We expand on NGR and CCR by introducing a parametric dependence of the forecast errors on forecast lead time and

long-term time trends hereafter named decadal climate forecast recalibration strategy (*DeFoReSt*). To better understand the properties of *DeFoReSt*, we conduct experiments using a toy model to produce synthetic forecast observation pairs with known properties. We compare the decadal recalibration with the drift correction proposed by Kruschke et al. (2015) to illustrate its benefits and limitations.

The remainder of the paper is organized as follows. In Sec. 2 we introduce the MiKlip decadal climate prediction system

and the corresponding reference data used. Moreover, we discuss how forecast quality of probabilistic forecasts is assessed. In Sec. 3, we motivate the extension of the NGR method named *DeFoReSt* and illustrate how verification calibration can be linked by the way the calibration parameters are estimated. The toy model used to study *DeFoReSt* is introduced and assessed in Sec. 4.2. In the following section, we apply the drift correction and *DeFoReSt* to decadal surface temperature predictions from the MiKlip system (Sec. 5). We assess global mean surface temperature and temperature over the North Atlantic subpolar gyre

region (60°-10°W, 50°-65°N). The investigated North Atlantic region has been identified as a key region for decadal climate



predictions with forecast skill for different parameters (e.g. Pohlmann et al., 2009; van Oldenborgh et al., 2010; Matei et al., 2012; Mueller et al., 2012). The paper closes with a discussion in Sec. 6.

## 2 Data and methods

### 2.1 Decadal climate forecasts

In this study we use retrospective forecasts (hereafter call hindcast) of surface temperature performed with the Max-Planck-Institute Earth System Model in a low-resolution configuration (MPI-ESM-LR). The atmospheric component of the coupled model is ECHAM6 run at a horizontal resolution of T63 with 47 vertical levels up to 0.1 hPa (Stevens et al., 2013). The ocean component is MPIOM with a nominal resolution of 1.58 and 40 vertical levels (Jungclaus et al., 2013).

We investigate one set of decadal hindcasts, namely from the MiKlip Prototype system, which consists 41 hindcasts, each
with 15 ensemble members, yearly initialized at January $1^{st}$ between 1961 and 2000 and then integrated for 10 years. The initialization of the atmospheric part was realized by full field initialization from fields of Era40 (Uppala et al., 2005) and Era-Interim (Dee et al., 2011), while the oceanic part was initialized with full fields from GECCO2 reanalysis (Köhl, 2015). Here, the full field initialization nudges the atmospheric or oceanic fields from the corresponding reanalysis to the MPI-ESM as full fields and not as anomalies. A detailed description of the Prototype system is given in Kröger et al. (2017, submitted).

### 2.2 Validation data

This study uses the 20th Century Reanalysis (20CR, Compo et al., 2011) for evaluation of the hindcasts. The reanalysis has been built by solely assimilating surface pressure observations, whereas the lower boundary forcing is given from HadISST1.1 sea surface temperatures and sea ice (Rayner et al., 2003). Moreover, 20CR is based on Ensemble-Kalman-filtering with 56 members and therefore also addresses observation and assimilation uncertainties. Additionally, 20CR covers the whole period
of the investigated decadal hindcasts, which is a major benefit over other common reanalysis data sets.

### 2.3 Assessing reliability and sharpness

Calibration or reliability refers to the statistical consistency between the forecast PDFs and the verifying observations. Hence, it is a joint property of the predictions and the verification. A forecast is reliable if forecast probabilities correspond to observed frequencies on average. Alternatively, a necessary condition for forecasts to be reliable is given if the time mean intra-ensemble
variance equals the mean squared error (MSE) between ensemble mean and observation (Palmer et al., 2006).

A common tool to evaluate the reliability and therefore the effect of a calibration is the rank histogram or *Talagrand diagram* which were separately proposed by Anderson (1996); Talagrand et al. (1997); Hamill and Colucci (1997). For a detailed understanding, the rank histogram has to be evaluated by visual inspection. Here, we have chosen to use the *Ensemble Spread Score* (ESS) as a summarizing measure. The ESS is the ratio between the time mean intra-ensemble variance $\bar{\sigma}^2$ and the mean





squared error between ensemble mean and observation, $MSE(\mu, y)$ (Palmer et al., 2006; Keller and Hense, 2011):

$$ESS = \frac{\bar{\sigma^2}}{MSE(\mu, y)},$$ (1)

with

$$\bar{\sigma^2} = \frac{1}{k}\sum_{j=1}^{k}\sigma_j^2,$$ (2)

and

$$MSE(\mu, y) = \frac{1}{k}\sum_{j=1}^{k}(y_j - \mu_j)^2.$$ (3)

Here, $\sigma_j^2, \mu_j$ and $y_j$ are the ensemble variance, the ensemble mean and the corresponding observation at time step $j$, with $j = 1, ..., k$, where $k$ is the number time steps.

Following Palmer et al. (2006), $ESS = 1$ indicates perfect reliability. The forecast is overconfident when $ESS < 1$, i.e. the
ensemble spread underestimates forecast error. If the ensemble spread is greater than the model error ($ESS > 1$), the forecast is overdispersive and the forecast spread overestimates forecast error. To better understand the components of the $ESS$, we also analyze the mean squared error $MSE$ of the forecast separately.

Sharpness, on the other hand, refers to the concentration or spread of a probabilistic forecast and is a property of the forecast only. A forecast is sharp, when it is taking a risk, i.e. when it is frequently different from the climatology. The
smaller the forecast spread, the sharper the forecast. Sharpness is indicative of forecast performance for calibrated and thus reliable forecasts, as forecast uncertainty reduces with increasing sharpness (subject to calibration). To assess sharpness, we use properties of the width of prediction intervals as in Gneiting and Raftery (2007). In this study the time mean intra-ensemble variance $\bar{\sigma^2}$ is used to asses the prediction width.

Scoring rules, finally, assign numerical scores to probabilistic forecasts and form attractive summary measures of predictive
performance, since they address reliability and sharpness simultaneously (Gneiting et al., 2005; Gneiting and Raftery, 2007; Gneiting and Katzfusss, 2014). These scores are generally taken as penalties, thus the forecasters seek to minimize them. A scoring rule is called proper, if its expected value is minimized when the observation is drawn from the same distribution as the predictive distribution. If a scoring rule is not proper, it is possible to minimize its expected value by predicting an unrealistic probability of occurrence. In simple terms, a forecaster would be rewarded for not being honest. Moreover a proper scoring
rule is called strictly proper if the minimum is unique. In this regard, the *Continuous Ranked Probability Score* ($CRPS$) is a suitable, strictly proper scoring rule for ensemble forecasts.

Given, $F$ is the predictive cumulative distribution function (CDF) and o is the verifying observation, the CRPS is defined as

$$CRPS(F, o) = \int_{-\infty}^{\infty}(F(y) - F_0(y))^2 dy,$$ (4)

where $F_0(y)$ is the Heaviside function and takes the values 0 or 1 if y is less than or greater equal than the observed value
$o$. Under the assumption that the predictive CDF is a normal distribution with mean $\mu$ and variance $\sigma^2$ Gneiting et al. (2005)



showed that (4) can be written as

$$
CRPS(\mathcal{N}(\mu,\sigma^2),o) =
$$
$$
\sigma\left\{\frac{o-\mu}{\sigma}[2\Phi\left(\frac{o-\mu}{\sigma}\right)-1]+2\varphi\left(\frac{o-\mu}{\sigma}\right)-\frac{1}{\sqrt{\pi}}\right\},
\tag{5}
$$

where $\Phi(\cdot)$ and $\varphi(\cdot)$ denote the CDF and the PDF, respectively, of the standard normal distribution.

The CRPS is negatively oriented. A lower CRPS indicates more accurate forecasts; a CRPS of zero denotes a perfect

(deterministic) forecast. Moreover, the average score over $k$ pairs of forecasts $F_j$ and observations $y_j$

$$
\overline{\mathrm{CRPS}} = \frac{1}{k}\sum_{j=1}^{k}CRPS(F_j,y_j)
\tag{6}
$$

reduces to the Mean Absolute Error ($MAE = \frac{1}{k}\sum_{j=1}^{k}|y_j-\mu_j|$) for deterministic forecasts (Gneiting and Raftery, 2004), i.e. $F_i$ in Eq. 6 would also be a step function. The CRPS can therefore be interpreted as a distance measure between the probabilistic forecast and the verifying observation (Siegert et al., 2015).

The Continuous Ranked Probability Skill Score ($CRPSS$) is, as the name implies, the corresponding skill score. A skill score relates the accuracy of the prediction system to the accuracy of a reference prediction (e.g. climatology). Thus, with a given $CRPS_F$ for the hindcast distribution and a given $CRPS_R$ for the reference distribution the $CRPSS$ can be defined as:

$$
CRPSS = 1-\frac{CRPS_F}{CRPS_R}.
\tag{7}
$$

Positive values of the CRPSS imply that the prediction system outperforms the reference prediction. Furthermore, this skill

score is unbounded for negative values (because hindcasts can be arbitrarily bad) but bounded by 1 for a perfect forecast.

## 3  *DeFoReSt*: Decadal Climate Forecast Recalibration Strategy

In the following paragraphs we discuss the decadal climate forecast recalibration strategy (*DeFoReSt*) and illustrate how forecast quality is used to estimate the parameters of the recalibration method.

We assume that the recalibrated predictive PDF $f^{Cal}(X|t,\tau)$ for random variable $X$ is a normal PDF with mean and variance

being functions of ensemble mean $\mu(t,\tau)$ and variance $\sigma^2(t,\tau)$, as well as start time $t$ and lead year $\tau$

$$
f^{Cal}(X|t,\tau) \sim \mathcal{N}(\alpha(t,\tau)+\beta(t,\tau)\mu(t,\tau),\gamma(t,\tau)^2\sigma^2(t,\tau)).
\tag{8}
$$

The term $\alpha(t,\tau)$ accounts for the mean or unconditional bias depending on lead year (i.e. the drift). Analogously $\beta(t,\tau)$ accounts for the conditional bias. Thus, the expectation $E[X] = \alpha(t,\tau)+\beta(t,\tau)\mu(t,\tau)$ could be a conditional and unconditional bias and drift adjusted deterministic forecast (we call a deterministic forecast a forecast without specifying uncertainty). For

now, we assume that the ensemble spread $\sigma(t,\tau)$ is sufficiently well related to forecast uncertainty such that it can be adjusted simply by a multiplicative term $\gamma(t,\tau)^2$. We thus refrain from using the additive term suggested for NGR by Gneiting et al.



(2005) to not end up with a too complex model, as the additive term should consequently be also a function of start time $t$ and lead time $\tau$; this term might be included in a future variant.

In the following, we motivate and develop linear parametric functions for $\alpha(t,\tau)$, $\beta(t,\tau)$ and $\gamma(t,\tau)$.

### 3.1 Addressing bias and drift: $\alpha(t,\tau)$

For bias and drift correction, we start with a parametric approach based on the studies of Kharin et al. (2012) and Kruschke et al. (2015). In their study, a third order polynomial captures the drift along lead time $\tau$ (Gangstø et al., 2013; Kruschke et al., 2015); the drift corrected forecasts $\hat{H}_{h,\tau,i}$ is approximated with a linear function of the forecast $H_{h,\tau,i}$ as

$$\hat{H}_{t,\tau,i} = H_{t,\tau,i} - (a_0 + a_1 t) - (a_2 + a_3 t)\tau - (a_4 + a_5 t)\tau^2 - (a_6 + a_7 t)\tau^3. \tag{9}$$

Here, $H_{t,\tau,i}$, is the raw, i.e. uncorrected, hindcast for the start time $t$, ensemble member $i$ and lead year $\tau$. In case the obser-
vations and model climatology have different climate trends, the bias between model and observations is non-stationary. Thus, the approach (9) also accounts for the dependency of the bias on the start year and therefore corrects errors in time trends. The parameters $a_0, ..., a_7$ are estimated by standard least-squares using the differences between the ensemble mean of all available hindcasts and the reanalysis, corresponding to the given start and lead time (Kruschke et al., 2015).

This motivates the following functional form for $\alpha(t,\tau)$ analogously to Eq. (9)

$$\alpha(t,\tau) = \sum_{l=0}^{3}(a_{2l} + a_{(2l+1)}t)\tau^l. \tag{10}$$

In principle arbitrary orders are possible for $t$ and $\tau$ as long as there is sufficient data to estimate the parameters.

### 3.2 Addressing conditional bias and ensemble spread: $\beta(t,\tau)$ and $\gamma(t,\tau)$

Additionally to adjusting the unconditional lead year dependent bias, *DeFoReSt* aims at simultaneously adjusting conditional
bias and ensemble spread. As a first approach, we take the same functional form for $\beta(t,\tau)$ and $\gamma(t,\tau)$:

$$\beta(t,\tau) = \sum_{l=0}^{3}(b_{2l} + b_{(2l+1)}t)\tau^l, \tag{11}$$

$$\gamma(t,\tau) = \log(\sum_{l=0}^{2}(c_{2l} + c_{(2l+1)}t)\tau^l). \tag{12}$$

The ensemble inflation $\gamma(t,\tau)$ is, however, assumed to be quadratic at most and constrained to be greater zero by using a static
logarithmic link function.

These assumption on model complexity are supported only by our experience; however, they remain subjective. A more transparent order selection will be topic of future work.





### 3.3 Parameter estimation

The coefficients $\alpha(t,\tau), \beta(t,\tau)$ and $\gamma(t,\tau)$ are now expressed as parametric functions of $t$ and $\tau$. The parameters are estimated by minimizing the average $CRPS$ over the training period (Gneiting et al., 2005). The associated score function is

$$\Gamma(\mathcal{N}(\alpha(t,\tau) + \beta(t,\tau)\mu, \gamma(t,\tau)^2\sigma^2), o) = \overline{\mathrm{CRPS}} =$$

$$\frac{1}{k}\sum_{j=1}^{k}\sqrt{\gamma(t,\tau)^2\sigma_j^2}\left\{ Z_j[2\Phi(Z_j) - 1] + 2\varphi(Z_j) - \frac{1}{\sqrt{\pi}} \right\}, \tag{13}$$

where

$$Z_j = \frac{O_j - (\alpha(t,\tau) + \beta(t,\tau)\mu_j)}{\sqrt{\gamma(t,\tau)^2\sigma_j^2}} \tag{14}$$

is the standardized forecast error for the $j$th forecast in the training data set. In the present study, optimization is carried out using the algorithm of Nelder and Mead (1965) as implemented in R (R Core Team, 2016).

   The initial guesses for optimization need to be carefully chosen to avoid local minima. Here, we obtain the $a_i$ and $b_j$ from

linearly modelling the observations $o$ with the forecast ensemble mean $\mu$, $t$ and $\tau$

$$o \sim \sum_{l=0}^{A}(a_{2l} + a_{(2l+1)}t)\tau^l + \sum_{l=0}^{B}(b_{2l} + b_{(2l+1)}t)\tau^l\mu, \tag{15}$$

using the notation for linear models from McCullagh and Nelder (1989); $c_0, c_1, c_2$ are set to zero which yields unity inflation $(\exp(\gamma(t,\tau)) = 1)$. However, convergence to a global minimum cannot be ensured.

## 4   Calibrating a toy model for decadal climate predictions

In this section, we apply *DeFoReSt* to a stochastic toy model, which is motivated from Weigel et al. (2009), but has been significantly altered to suit the needs of this study. Here, a detailed description of the toy models construction is given in the following subsection. Subsequently, we assess *DeFoReSt* for two exemplary toy model setups.

### 4.1   Toy model construction

The toy model consists of two parts which are detailed in the following two subsections: a) Pseudo-observations, the part

generating a substitute $x(t+\tau)$ for the observations, and b) Pseudo-forecasts, the second part deriving an associated ensemble prediction $f(t,\tau)$ from this observations. The third subsection motivates the choice of parameters for the toy model.

### 4.1.1   Pseudo-observations

We construct a toy model setup simulating ensemble predictions for the decadal time scale and associated pseudo-observations. Both are based on an arbitrary but predictable signal $\mu_x$. The pseudo-observations $x$ (e.g. annual means of surface temperature



over a given area) is the sum of this predictable signal $\mu_x$ and an unpredictable noise term $\epsilon_x$,

$$x(t+\tau) = \mu_x(t+\tau) + \epsilon_x(t+\tau). \tag{16}$$

Following Kharin et al. (2012) $\mu_x$ can be interpreted as the atmospheric response to slowly varying and predictable boundary conditions, while $\epsilon_x$ represents the unpredictable chaotic components of the observed dynamical system. $\mu_x$ and $\epsilon_x$ are assumed to be stochastic Gaussian processes

$$\mu_x(t+\tau) \sim \mathcal{N}(0, \sigma_{\mu_x}^2) \qquad \text{with} \qquad \sigma_{\mu_x}^2 = \eta^2 \leq 1 \tag{17}$$

and

$$\epsilon_x(t+\tau) \sim \mathcal{N}(0, \sigma_{\epsilon_x}^2) \qquad \text{with} \qquad \sigma_{\epsilon_x}^2 = 1 - \eta^2. \tag{18}$$

The variation of $\mu_x$ around a slowly varying climate signal can be interpreted as the predictable part of decadal variability, its amplitude is given by the variance $\text{var}(\mu_x(t+\tau)) = \sigma_{\mu_x}^2$. The total variance of the pseudo-observations is thus $\text{Var}(x) = \sigma_x^2 = \sigma_{\mu_x}^2 + \sigma_{\epsilon_x}^2$. Here, the relation of the latter two is uniquely controlled by the parameter $\eta \in [0,1]$, which can be interpreted as potential predictability ($\eta^2 = \sigma_{\mu_x}^2 / \sigma_x^2$).

In this toy model setup, the concrete form of this variability is not considered and thus taken as random. A potential climate trend could be superimposed as a time varying mean $\mu(t) = E[x(t)]$. As for the recalibration strategy only a difference in trends is important, we use $\mu(t) = 0$ and $\alpha(t,\tau)$ addressing this difference in trends of forecast and observations.

### 4.1.2 Pseudo-forecasts

We now specify a model giving a potential ensemble forecast with ensemble members $f_i(t,\tau)$ for observations $x(t+\tau)$:

$$f_i(t,\tau) = \mu_{ens}(t,\tau) + \epsilon_i(t,\tau), \tag{19}$$

where $\mu_{ens}(t,\tau)$ is the ensemble mean and

$$\epsilon_i(t,\tau) \sim \mathcal{N}(0, \sigma_{\text{ens}}^2(t,\tau)) \tag{20}$$

is the deviation of ensemble member $i$ from the ensemble mean; $\sigma_{\text{ens}}^2$ is the ensemble variance. In general, ensemble mean and ensemble variance both can dependent on lead time $\tau$ and start time $t$. We relate the ensemble mean $\mu_{ens}(t,\tau)$ to the predictable signal in the observations $\mu_x(t,\tau)$ by assuming a) a systematic deviation characterized by an unconditional bias $\chi(t,\tau)$ (accounting also for a drift and difference in climate trends), a conditional bias $\psi(t,\tau)$ and b) a random deviation $\epsilon(t,\tau)$:

$$\mu_{ens}(t,\tau) = \chi(t,\tau) + \psi(t,\tau) \left( \mu_x(t,\tau) + \epsilon_f(t,\tau) \right), \tag{21}$$

with $\epsilon_f(t,\tau)) \sim \mathcal{N}(0, \sigma_{\epsilon_f}(t,\tau))$ being a random forecast error with variance $\sigma_{\epsilon_f}^2(t,\tau) < \sigma_{\epsilon_x}^2$. Although the variance of the random forecast error can in principle be dependent on lead time $\tau$ and start time $t$, we assume for simplicity a constant variance $\sigma_{\epsilon_f}^2(t,\tau) = \sigma_{\epsilon_f}^2$.





We further assume an ensemble dispersion related to the variability of the unpredictable noise term $\epsilon_x$ with an inflation factor $\omega(t, \tau)$

$$\sigma_{ens}^2(t, \tau) = \omega^2(t, \tau) \left( \sigma_{\epsilon_x}^2 - \sigma_{\epsilon_f}^2 \right). \tag{22}$$

According to Eq. 21 the forecast ensemble mean $\mu_{\mathrm{ens}}$ is simply a function of the predictable signal $\mu_x$. In this toy model

formulation, an explicit formulation of $\mu_x$ is not required, hence a random signal might be used for simplicity and it would be legitimate to assume $E[\mu_x] = \mu(t + \tau) = 0$ without restricting generality. Here, we propose a linear trend in time $E[\mu_x] = \mu(t + \tau) = m_0 + m_1 t$ to emphasize a typical problem encountered in decadal climate prediction: different trends in observations and predictions (Kruschke et al., 2015).

### 4.1.3 Choosing the toy models' parameters

This toy model setup is controlled by four parameters: The first parameter $\eta$ determines the ratio between the variances of the predictable signal and the unpredictable noise term (and thus characterizes potential predictability, see Sec. 4.1.2). Here, we investigate two cases: one with low ($\eta = 0.2$) and one with high potential predictability ($\eta = 0.8$).

The remaining three parameters are $\chi(t, \tau)$, $\psi(t, \tau)$ and $\omega(t, \tau)$, which control the unconditional and the conditional bias and the dispersion of the ensemble spread. To have a toy model experiment related to observations, $\chi(t, \tau)$ and $\psi(t, \tau)$ are based

on the correction parameters obtained from calibrating the MiKlip Prototype ensemble surface temperature over the North Atlantic against NCEP 20CR reanalyses; $\chi(t, \tau)$ and $\psi(t, \tau)$ are based on ratios of polynomials up to 3rd order (in lead years), Eqs. A1 and A2) with coefficients varying with start years (see Figs. 1a and 1b).

The ensemble inflation factor $\omega(t, \tau)$ is chosen such that the forecast is overconfident for the first lead years and becoming underconfident later; this effect intensifies with start years, see Fig. 1c. A more detailed explanation and numerical values used

for the construction of $\chi(t, \tau)$, $\psi(t, \tau)$ and $\omega(t, \tau)$ are given in Appendix A.

Given this setup, a choice of $\chi(t, \tau) \equiv 0$, $\psi(t, \tau) \equiv 1$ and $\omega(t, \tau) \equiv 1$ would yield a perfectly calibrated ensemble forecast:

$$f^{\mathrm{perf}}(t, \tau) \sim \mathcal{N}(\mu_x(t, \tau), \sigma_{\epsilon_x}^2(t, \tau)). \tag{23}$$

The ensemble mean $\mu_x(t, \tau)$ of $f^{\mathrm{perf}}(t, \tau)$ is equal to the predictable signal of the pseudo-observations. The ensemble variance $\sigma_{\epsilon_x}^2(t, \tau)$ is equal to the variance of the unpredictable noise term representing the error between the ensemble mean of $f^{\mathrm{perf}}(t, \tau)$

and the pseudo-observations. Hence, $f^{\mathrm{perf}}(t, \tau)$ is perfectly reliable.

Analogous to the MiKlip experiment, the toy model uses 50 start years ($t = 0, \ldots, 49$), each with 10 lead years $\tau = 1, \ldots, 10$, and 15 ensemble members ($i = 1, \ldots, 15$). The corresponding pseudo-observations $x(t + \tau)$ run over a period of 59 years in order to cover lead year 10 of start year 50.

### 4.2 Toy model verification

To assess *DeFoReSt* we consider two extreme toy model setups. The two setups are designed such that the predictable signal is stronger than the unpredictable noise for higher potential predictability (setup 1) and vice versa (setup 2, cf. Sec. 4.1). For




each toy model setup we calculated the $ESS$, the $MSE$, time mean intra-ensemble variance and the $CRPSS$ with respect to climatology for the corresponding recalibrated toy model.

In addition to the recalibrated pseudo-forecast, we compare

- a *raw* pseudo-forecast (no correction of unconditional, conditional bias and spread),

- a *drift-corrected* pseudo-forecast (no correction of conditional bias and spread), and

- a *perfect* pseudo-forecast (Eq.23, available only in this toy model setup)

All scores have been calculated using cross-validation with a yearly moving calibration window with a width of 10 years. A detailed description of this procedure is given in appendix B.

The $CRPSS$ and reliability values of the perfect forecast could be interpreted as optimum performance within the associated

toy model setup, due to the missing bias and ensemble dispersion. For instance, the perfect model's $CRPSS$ with respect to climatology would be 1 for a toy model setup with perfect potential predictability ($\eta = 1$) and zero for a setup with no potential predictability ($\eta = 0$). Hence, the climatology could not be outperformed by any prediction model when no predictable signal is existing.

### 4.2.1   A toy model setup with high potential predictability

Figures 2a and 2c show the temporal evolution of the toy model data before and after recalibration with *DeFoReSt* together with the corresponding pseudo-observations. Before recalibration, the pseudo-forecast apparently exhibits the characteristic problems of a decadal ensemble prediction: unconditional bias (drift), conditional bias and underdispersion, which are lead and start time dependent. Additionally, the pseudo-observations and the pseudo-forecast have different trends. After recalibration, the lead and start time dependent biases are corrected, such that the temporal evolution of the pseudo-observations is mostly

represented by the pseudo-forecast.

Moreover, the pseudo-forecast is almost perfectly reliable after recalibration (not underdispersive), which could be shown with the $ESS$ (Fig. 3a). Here, the recalibrated model is nearly identical to the perfect model for all lead years with reliability values close to 1.

The recalibrated forecast outperforms the raw model output and the drift corrected forecast, which $ESS$ values are lower one

and thus underdispersive. The reduced performance of the raw models and the drift correction is a result of the toy model design, yielding to a higher ensemble mean variance combined with a decreased ensemble spread. In addition, the increased variance of the ensemble mean also results in an increased influence of the conditional bias. The problem is, the raw model forecast and the drift correction could not account for that conditional bias, because neither the ensemble mean nor the ensemble spread were corrected by these forecasts. Therefore, the influence of the conditional bias also becomes noticeable for the reliability of

the raw model and the drift corrected forecast; one can see that the minimum and maximum of the conditional bias (see Fig. 1) is reproduced by the reliability values of these forecasts.

Regarding the differences between raw model and the drift corrected forecast, it is visible that the latter outperforms the raw model. The explanation is that the drift correction accounts for the unconditional bias, while the raw model does not correct





this type of error. Here, one can see the impact of the unconditional bias on the raw model. Nonetheless, the influence of the unconditional bias is rather small, compared to the conditional bias.

The effect of unconditional and conditional bias is illustrated in Fig. 3b), which shows the $MSE$ of the different forecasts to the pseudo observations. Here, the drift corrected forecast outperforms the raw model. These forecasts are outperformed by

the recalibrated forecast, which simultaneously corrects the unconditional and conditional bias. In this regard, both biases are corrected properly because the $MSE$ of the recalibrated forecast is almost equal to the perfect models $MSE$.

The sharpness of the different forecasts are compared by calculating the time mean intra-ensemble variance (see Fig. 3c). For all lead years, the raw model and the drift corrected forecast exhibit the same sharpness, because the ensemble spread is unmodified for both forecasts.

Another notable aspect is that the raw and drift corrected forecast have a higher sharpness (i.e. lower ensemble variance) than the perfect model for lead years 1 to 4 and vice versa for lead years 5 to 10. This is due to the toy models incorporated underdispersion for the first lead years and an overdispersion for later lead years. Therefore the sharpness of the perfect model could be interpreted as the maximum sharpness of the model without being unreliable.

The sharpness of the recalibrated forecast is very similar to the sharpness of the perfect model for all lead years. The

recalibration therefore performs well in correcting under- and overdispersion in the toy model forecasts.

A joint measure for sharpness and reliability is the $CRPS$ and consequently the $CRPSS$ with respect to climatology, where the latter is shown in Fig. 3d. The relatively low $CRPSS$ values of the raw and drift corrected forecast are mainly affected by their reliability; i.e. the unconditional and conditional bias influences are also noticeable for this skill score. Thus, both models exhibit a maximum at lead year 2 and a minimum at lead year 7, where the drift corrected forecast performs better. However,

the raw model and the drift corrected forecast are inferior to climatology (the $CRPSS$ is below zero) for all lead years.

In contrast, the recalibrated forecast approaches $CRPSS$ values around 0.5 for all lead years and performs nearly identical to the perfect model. This illustrates that the unconditional bias, conditional bias and ensemble dispersion can be corrected with this method.

### 4.2.2 A toy model setup with low potential predictability

Figures 2b and 2d show the temporal evolution of the toy model data with a low potential predictability before and after recalibration with *DeFoReSt* together with the corresponding pseudo-observations. Before recalibration, the pseudo-forecast is underdispersive for the first lead years, whereas the ensemble spread increases for later lead years. Moreover, the pseudo-forecast exhibits lead and start time dependent (unconditional) bias (drift) and conditional bias.

After recalibration, the lead and start time dependent biases are corrected, such that the recalibrated forecast mostly describes

the trend of the pseudo-observations.

The recalibrated forecast is also reliable (Fig. 4a); it performs as well as the the perfect model. Here, the value of the $ESS$ is close to one for both forecasts. Thus, comparing the reliability of the setups with low and high potential predictability, no differences are recognizable. The reason is, the ratio between $MSE$ and ensemble variance, characterizing the $ESS$, does





not change much; the worse $MSE$ performance of the recalibrated forecast (Fig. 4b) is compensated with a higher ensemble variance (Fig. 4c).

On the contrary, one can see a general improvement of the raw and drift corrected forecasts' reliability compared to the model setup with high potential predictability. The reason is, that the low potential predictability $\eta$ of this toy model setup yields to

smaller variance of the ensemble mean, i.e. the conditional bias has a minor effect. Another aspect for the comparatively good performance of the raw model, is the increased ensemble spread, yielding to an enhanced representation of the unconditional bias.

The minor effect of the conditional bias within the low potential predictability setup is also represented by the $MSE$ (Fig. 4b). Here, the difference between drift corrected and recalibrated forecast has decreased w.r.t. the high potential pre-

dictability setup. Comparing both toy model setups, it is also apparent that, for a setup with $\eta = 0.2$, the $MSE$ generally has increased for all forecasts. The reason is, that the predictable signal decreases for a lower $\eta$. Therefore, even the perfect models $MSE$ has increased.

Figure 4c shows the time mean intra-ensemble variance for the toy model setup with low potential predictability. It is notable that the ensemble variance for this low potential predictability setup is generally greater than for a high $\eta$ (Fig. 3c). This is due

to the fact that the total variance in the toy model is constrained to one and a lower $\eta$ therefore yields to a greater ensemble spread.

Nonetheless, the raw model and drift corrected forecast also still have a higher sharpness (i.e. lower ensemble variance) than the perfect model for lead years 1 to 4 and vice versa for lead years 5 to 10. Here, the reason for this is again the construction of the toy model, with an underdispersion for the first lead years and an overdispersion for later lead years.

The recalibrated forecast reproduces the perfect models sharpness also quite well for the potential predictability setup.

Figure 4d shows the $CRPSS$ with respect to climatology. Firstly, it is apparent that the weak predictable signal of this toy model setup shifted the $CRPSS$ of all models closer to zero or the climatological skill. Nevertheless, please note that the recalibrated forecast is almost as good as the perfect model and that it is slightly superior to the drift corrected forecast. We conclude that the recalibration works well also in situations with limited predictability.

## 25  5  Calibrating decadal climate surface temperature forecasts

While in Sec. 4.2 *DeFoReSt* was applied to toy model data, in this section *DeFoReSt* will be applied on surface temperature of MiKlip Prototype runs with MPI-ESM-LR. Here, global mean and a spatial mean values over the North Atlantic subpolar gyre (60°-10°W, 50°-65°N) region will be analyzed.

Analogous to the previous section we compute the $ESS$, the $MSE$ the intra-ensemble variance and the $CRPSS$ with

respect to climatology. In this section, a 95% confidence interval was additionally calculated for these metrics using a bootstrapping approach with 1000 replicates. Furthermore, all scores have been calculated using cross-validation with a yearly moving calibration window with a width of 10 years (see appendix B).





## 5.1 North Atlantic mean surface temperature

Figures 5a and 5b show the temporal evolution of North Atlantic mean surface temperature before and after recalibration with the corresponding NCEP 20CR reference. Before recalibration, the MiKlip Prototype hindcasts exhibit a lead time dependent bias (drift) and a lead time dependent ensemble spread. Here, lead time dependent bias of Prototype is a consequence of an initialization shock due to a full-field initialization (Meehl et al., 2014; Kruschke et al., 2015; Kröger et al., 2017, submitted). After recalibration with *DeFoReSt* the drift of the MiKlip Prototype was corrected and the ensemble spread is also modified.

Regarding the reliability, Fig. 6a shows the $ESS$. The recalibrated forecast is almost perfectly reliable for all lead years because all $ESS$ values of this model are close to one. Moreover, the recalibrated forecast is more skillful than the drift corrected forecast for years 3 to 10, where the improvement is only significant for lead year 4 to 8. It is also apparent that the drift corrected forecast is significantly overdispersive for lead years 3 to 10. For lead years 1 and 2 both post processing methods perform equally well. On the contrary, the raw model's reliability is obviously inferior to the post processed models and significantly underdispersive for all lead years. This implies that the unconditional bias induces most of the systematic error of the MiKlip Prototype runs.

Regarding the $MSE$, one can see that the recalibrated forecast outperforms the drift corrected forecast for lead years 1 and 2 and 8 to 10 (Fig. 6b). Although this improvement of the recalibrated forecast is not significant, it may be still attributed to its correction of the conditional bias. Here, the raw model performs obviously worse compared to the post processed models, because neither the unconditional nor the conditional bias were corrected.

Figure 6c shows the spread as measured by the time mean intra-ensemble variance for the North Atlantic mean surface temperature. The ensemble variance of the raw model and the drift corrected forecast is equal, since the ensemble spread of the drift corrected forecast was not corrected. Here, the ensemble variance of both models is increasing with lead times. The ensemble variance of the recalibrated forecast is lower than the variance of the raw and drift corrected forecast for the first lead years 2 to 10, i.e. the recalibrated forecast has a higher sharpness than the other two forecasts. The combination of increasing ensemble variance and almost constant $MSE$ yielding to the identified increasing underconfidence (see Fig. 6a) of the drift corrected forecast for that period.

Figure 6d shows that in terms of $CRPSS$ both the drift corrected forecast and the recalibrated forecast outperform the raw model. Here, the $CRPSS$ of the raw model is smaller than -1 for all lead year, thus the corresponding graph lies below the plotted range. *DeFoReSt* slightly performs better (but not significantly better) than the drift corrected forecast for almost all lead years, except lead year 3 and 4. Additionally, the $CRPSS$ with respect to climatology shows that the recalibrated forecast outperforms a constant climatological forecast for all lead times and is significantly better for lead years 1 and 3 to 10.

## 5.2 Global mean surface temperature

Figures 7a and 7b show the temporal evolution of global mean surface temperature before (see equation 19) and after recalibration with the corresponding NCEP 20CR reference. Before recalibration with *DeFoReSt*, the MiKlip Prototype hindcasts exhibit a lead time dependent bias (drift) and a lead time dependent ensemble spread. The drift of the global mean surface





temperature is even stronger than the North Atlantic counterpart. After applying *DeFoReSt*, the drift of the MiKlip Prototype was corrected and the ensemble spread is fairly constant for all lead times.

The $ESS$ for a global mean surface temperature is shown in Fig. 8a. It can be seen that the recalibrated forecast is also perfectly reliable for the global mean surface temperature. Here, all $ESS$ values are near one. Additionally, the recalibrated

forecast is more skillful than the drift corrected forecast for all lead years. Here, only lead year 1 and 10 are significant. The reliability values of the drift corrected forecast indicate a significantly overconfidence for almost every lead year. As for the North Atlantic mean, the raw model's reliability for a global mean temperature is inferior to the post processed models.

Figure 8b shows the $MSE$. It is apparent that the recalibrated forecast outperforms the drift corrected forecast for all lead years, where the improvement for lead years 5 to 6 and 8 to 10 is significant. Moreover, the $MSE$ of the drift corrected

forecast increases with lead years, while the $MSE$ of the recalibrated forecast is constant. Thus, this increasing difference between these forecasts is an effect of a lead year dependency of the conditional bias.

Figure 8c shows the time mean intra-ensemble variance for the global mean surface temperature. Regarding sharpness, the drift corrected and the recalibrated forecast perform equally for lead years 2 and 3. Hence, the improved reliability of the recalibrated forecast could not attributed to a modified ensemble spread. The explanation is that the recalibration method also

accounts for conditional and unconditional bias, while the drift correction method only addresses to the unconditional bias. Thus, the error between observation and ensemble mean of the recalibrated forecast is lower than the error of the drift corrected forecast (see Fig. 8b). Consequently, the drift corrected forecast is overconfident for this period (see Fig. 8a), due to a greater error combined with an equal sharpness.

Regarding the $CRPSS$, Fig. 8d shows that *DeFoReSt* performs significantly better than the drift corrected forecast for lead

years 1, and 8 to 10. Furthermore, the $CRPSS$ shows that these forecasts also outperforming the climatology, where the improvement of the drift corrected forecast against climatology is solely not significant for lead years 8 to 9. The $CRPSS$ of the raw model is smaller than -1 for all lead years and therefore out of the shown range.

All in all, the better $CRPSS$ performance of *DeFoReSt* model could be explained due to a superior reliability for all lead years (see Fig. 8a).

## 25  6   Summary and conclusions

There are many studies describing recalibration methods for weather and seasonal forecasts (e.g. Gneiting et al., 2005; Weigel et al., 2009). Regarding decadal climate forecasts, those methods cannot be applied easily, because decadal climate prediction systems on that time scale exhibit characteristic problems including model drift (lead time dependent unconditional bias) and climate trends which could differ from observations. In this regard Kruschke et al. (2015); Kharin et al. (2012) proposed

methods to account for lead and start time dependent unconditional biases of decadal climate predictions.

In addition to unconditional biases, probabilistic forecasts could show lead and start year dependent conditional biases and under- or overdispersion. Therefore, we proposed the post processing method *DeFoReSt* which accounts for the three above mentioned issues. Following the suggestion for the unconditional bias (Kruschke et al., 2015), we allow for the conditional




bias and the ensemble dispersion to change polynomially with lead time and linearly with start time. Two advantages of a polynomial fit over the common exponential fit (e.g., as proposed by Kharin et al. (2012)) are stated by Gangstø et al. (2013): First, for a small sample size (this is given for decadal climate predictions) the fit of an exponential with offset is relatively difficult and unreliable. Second, a polynomial approach can capture a local maximum/minimum of the above mentioned errors

at a specific lead time; the evolution of these errors may be non-monotonous. Analog to Kruschke et al. (2015), we chose a third order polynomial approach for the correction parameter of the unconditional bias and the conditional bias. A second order polynomial approach is chosen for the correction parameter of the ensemble dispersion. Note that these choices might influence the resulting forecast skill. It might be worth using a transparent model selection strategy, this is topic of future research. The associated *DeFoReSt* parameters are estimated by minimization of the $CRPS$ (Gneiting et al., 2005). The $CRPSS$, the $ESS$,

the time mean intra-ensemble variance (as measure for sharpness) and the $MSE$ assess the performance of *DeFoReSt*. All scores were calculated with 10 year block-wise cross-validation.

We investigated *DeFoReSt* using toy model simulations with high ($\eta = 0.8$) and low potential predictability ($\eta = 0.2$). Errors based on the same polynomial structure as used for the recalibration method were impose. *DeFoReSt* is compared to a conventional drift correction and a perfect toy model without unconditional bias, conditional bias and ensemble spread dispersion

was used as a benchmark. Here, the recalibration and drift correction benefits from the fact that the structure of errors imposed is known. Although the model for the error structure is flexible, the gain in skill is an upper limit to other applications where the structure of errors is unknown. Conclusions on the relative advantage of *DeFoReSt* over the drift correction for different potential predictability setups, however, should be largely unaffected by the choice of toy model errors.

A recalibrated forecast shows (almost) perfect reliability ($ESS = 1$). Sharpness can be improved due to the correction

of conditional and unconditional biases. Thus, given a high potential predictability ($\eta = 0.8$), recalibration leads to major improvements in skill ($CRPSS$) over a climatological forecast. Forecasts with low potential predictability ($\eta = 0.2$) improve also but the gain in skill ($CRPSS$) over a climatological forecast is limited. In both cases, reliability, sharpness and thus $CRPSS$ of the recalibrated model are almost equal to the perfect model. *DeFoReSt* outperforms the drift corrected forecast with respect to $CRPSS$, reliability and $MSE$, due to additional correction of the conditional bias and the ensemble dispersion.

The differences between these two post processed forecasts are, however, smaller for the low potential predictability setup.

We also applied *DeFoReSt* to surface temperature data of the MiKlip Prototype decadal climate forecasts, spatially averaged over the North Atlantic subpolar gyre region and a global mean. Pronounced predictability for these cases has been identified by previous studies (e.g. Pohlmann et al., 2009; van Oldenborgh et al., 2010; Matei et al., 2012; Mueller et al., 2012). Nonetheless, both regions are also affected by model drift (Kröger et al., 2017, submitted). The North Atlantic region shows an overconfident

forecasts for all lead years for the raw model output. The drift corrected forecast is underconfident for lead years 8 to 10. The recalibrated forecast is almost perfectly reliable for all lead years ($ESS = 1$) and outperforms the drift correction method with respect to $CRPSS$ for lead years 1 and 2 and 5 to 10. For the global mean surface temperature *DeFoReSt* significantly outperforms the drift corrected forecast for several lead years with respect to $CRPSS$. The $CRPSS$ in for the global case is generally higher than for the North Atlantic region. The recalibrated global forecast is perfectly reliable; the drift corrected

forecast, however, tends to be overconfident for all lead years. This is in accordance to other studies suggesting that ensemble



forecasts typically underestimate the true uncertainty and tend to be overconfident (Weigel et al., 2009; Hamill and Colucci, 1997; Eckel and Walters, 1998). *DeFoReSt* thus accounts for both, underdispersive and overdispersive forecasts.

*DeFoReSt* with third/second order polynomials is quite successful. However, it is worthwhile investigating the use of order selection strategies, such as LASSO (Tibshirani, 1996) or information criteria. Furthermore parameter uncertainty due to a

small training size may result in forecast that are still underdispersive after recalibration. For the seasonal scale, this has been discussed by Siegert et al. (2015). However, for decadal climate forecasts, this aspect should be further considered in future studies. Recalibration based on $CRPS$-minimization is computationally expensive which might become problematic if not regional means but individual grid points are considered. As an alternative to $CRPS$-minimization Vector Generalized Linear Model (VGLM, Yee, 2008) might be considered which have been implemented in an efficient way.

Based on simulations from a toy model and the MiKlip decadal climate forecast system we could show that *DeFoReSt* is a consistent recalibration strategy for decadal forecast leading to reliable forecast with increased sharpness due to simultaneous adjustment of conditional and unconditional biases depending no lead-time.

*Code and data availability.* The NCEP 20CR reanalysis used in this study are freely accessible through NCAR (National Center for Atmospheric Research) after a simple registration process (https://www.esrl.noaa.gov/psd/data/20thC_Rean). The MiKlip Prototype data used

for this paper are from the BMBF-funded project MiKlip and are available on request (https://www.fona-miklip.de). The post-processing (incl. cross-validated recalibration and drift correction) and toy model algorithms are implemented using GNU licensed free software from the R Project for Statistical Computing (http://www.r-project.org). Version 1.0 of these implementations is available under https://doi.org/10.5281/zenodo.831588.

## Appendix A: Construction of the toy model's free parameters

For this toy model setup, $\chi(t,\tau)$ and $\psi(t,\tau)$ are obtained from $\alpha(t,\tau)$ and $\beta(t,\tau)$ as follows:

$$\chi(t,\tau) = -\frac{\alpha(t,\tau)}{\beta(t,\tau)} \tag{A1}$$

$$\psi(t,\tau) = \frac{1}{\beta(t,\tau)} \tag{A2}$$

$$\omega(t,\tau) = \frac{1}{\gamma(t,\tau)}. \tag{A3}$$

The parameters $\chi(t,\tau)$, $\psi(t,\tau)$ and $\omega(t,\tau)$ are defined such that a perfectly recalibrated toy model forecast $f^{\text{Cal}}$ would have

the following form:

$$f_i^{\text{Cal}}(t,\tau) = \alpha(t,\tau) + \beta(t,\tau)\mu_{ens}(t,\tau) + \gamma(t,\tau)\epsilon_i, \tag{A4}$$

where $\epsilon_i$ is the deviation of each ensemble member $i$ from the ensemble mean $\mu_{ens}(t,\tau)$. Here, $\sigma_{\text{ens}}^2$ is the ensemble variance. Writing (A4) as Gaussian distribution and applying the definitions of $\mu_{\text{ens}}$ (Eq. 21) and $\sigma_{\text{ens}}$ (Eq. 22) leads to

$$f_i^{Cal}(t,\tau) \sim \mathcal{N}(\alpha(t,\tau) + \beta(t,\tau)(\chi(t,\tau) + \psi(t,\tau)\mu_x(t,\tau)), \gamma(t,\tau)\omega(t,\tau)\sigma_{\epsilon_x}^2(t,\tau)), \tag{A5}$$




and applying the definitions of $\chi(t,\tau)$, $\psi(t,\tau)$ and $\omega(t,\tau)$ (Eqs. A1-A3) to (A5) would further lead to:

$$f_i^{Cal}(t,\tau) \sim \mathcal{N}(\alpha(t,\tau) - \beta(t,\tau)\frac{\alpha(t,\tau)}{\beta(t,\tau)} + \frac{\beta(t,\tau)}{\beta(t,\tau)}\mu_x(t,\tau), \frac{\gamma(t,\tau)}{\gamma(t,\tau)}\sigma_{\epsilon_x}^2(t,\tau)), \tag{A6}$$

This shows that $f^{Cal}$ is equal to the perfect toy model $f^{Perf}(t,\tau)$ (Eq. 23):

$$f^{Cal}(t,\tau) \sim \mathcal{N}(\mu_x(t,\tau), \sigma_{\epsilon_x}^2(t,\tau)). \tag{A7}$$

This setting has the advantage that the perfect estimation of $\alpha(t,\tau)$, $\beta(t,\tau)$ and $\gamma(t,\tau)$ is already known prior to calibration with $CRPS$-minimization.

Following the suggestion of Kruschke et al. (2015), a third order polynomial approach was chosen for unconditional $\alpha(t,\tau)$ and conditional bias $\beta(t,\tau)$ as well as for the inflation factor $\omega(t,\tau)$, yielding

$$\alpha(t,\tau) = (a_0 + a_1 t) + (a_2 + a_3 t)\tau + (a_4 + a_5 t)\tau^2 + (a_6 + a_7 t)\tau^3, \tag{A8}$$

$$\beta(t,\tau) = (b_0 + b_1 t) + (b_2 + b_3 t)\tau + (b_4 + v_5 t)\tau^2 + (b_6 + b_7 t)\tau^3 \text{ and} \tag{A9}$$

$$\omega(t,\tau) = (w_0 + w_1 t) + (w_2 + w_3 t)\tau + (w_4 + w_5 t)\tau^2 + (w_6 + w_7 t)\tau^3. \tag{A10}$$

For the current toy model experiment, we exemplarily specify values for $u_i$ and $v_i$ as obtained from calibrating the ensemble mean of MiKlip Prototype GECCO2 ($\bar{f}_{Prot}$) surface temperature over the North Atlantic against NCEP 20CR reanalyses ($T_{obs}$):

$$E[T_{obs}] \sim (a_0 + a_1 t) + (a_2 + a_3 t)\tau + (a_4 + a_5 t)\tau^2 + (a_6 + a_7 t)\tau^3 +$$
$$((b_0 + b_1 t) + (b_2 + b_3 t)\tau + (b_4 + b_5 t)\tau^2 + (b_6 + b_7 t)\tau^3)\bar{f}_{Prot}. \tag{A11}$$

The values of the coefficients are given in Tab. A1 (upper and middle row). The last row of Tab. A1 gives the values of $w_i$, i.e. the series expansion of the inflation factor $\omega(t,\tau)$. These are chosen such that the forecast is overconfident for the first lead years and becoming underconfident for later lead years (see Fig. 1c).

**Appendix B: Cross-validation procedure for for decadal climate predictions**

We propose a cross-validation setting for decadal climate predictions to ensure fair conditions for assessing the benefit of a post processing method over a raw model without any post processing. All scores are calculated with a yearly moving validation period with a length of 10 years. This means that one start year including 10 lead years was left out for validation. The remaining start years and the corresponding lead years were used for estimating the correction parameters for the prediction 25  within the validation period; start years within the validation period were not taken into account. This procedure was repeated for a start year wise shifted validation period.

This setting is illustrated in Fig. A1 for an exemplary validation period from 1964 to 1973, i.e. the correction parameters are estimated for all hindcasts which are initialized outside the validation period (1962; 1963; 1974; 1975,...).





*Competing interests.* The authors declare that they have no conflict of interest.

*Acknowledgements.* This study was funded by the German Federal Ministry for Education and Research (BMBF) project MiKlip (sub-projects CALIBRATION Förderkennzeichen FKZ 01LP1520A and FLEXFORDEC Förderkennzeichen: FKZ 01LP1519A). Mark Liniger and Jonas Bend have received funding from the European Union's Seventh Framework Programme [FP7/2007- 2013] under Grant Agreement
5   no 308291.



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





**Figure 1.** Unconditional bias (a, $\chi(t,\tau)$), conditional bias (b, $\psi(t,\tau)$), and dispersion of the ensemble spread (c, $\omega(t,\tau)$) as a function of lead year $\tau$ with respect to different start years $t$.



**Figure 2.** Temporal evolution of the raw (a, b) and with *DeFoReSt* recalibrated (c, d) pseudo-forecast for different start years (colored lines) with potential predictability $\eta = 0.8$ (a, c) and $\eta = 0.2$ (b, d). Each pseudo-forecast runs over 10 lead years. The black line represents the associated pseudo-observation.



**Figure 3.** Reliability (a), MSE (b), Ensemble Variance (c) and CRPSS (d) of the raw toy model (black line), the drift corrected toy model forecast (red line), recalibrated (*DeFoReSt*) toy model forecast (blue line) and the perfect toy model (green line) for $\eta = 0.8$. The drift correction method does not account for the ensemble spread, thus the ensemble variance of the raw model and the drift corrected forecast are equal. For reasons of clarity the raw models CRPSS with values between -5 and -9 are not shown here.





**Figure 4.** Reliability (a), MSE (b), Ensemble Variance (c) and CRPSS (d) of the raw toy model (black line), the drift corrected toy model (red line), recalibrated (*DeFoReSt*) toy model (blue line) and the perfect toy model (green line) for $\eta = 0.2$. The drift correction method does not account for the ensemble spread, thus the ensemble variance of the raw model and the drift corrected forecast are equal. For reasons of clarity the raw models CRPSS with values between -3 and -9 are not shown here.




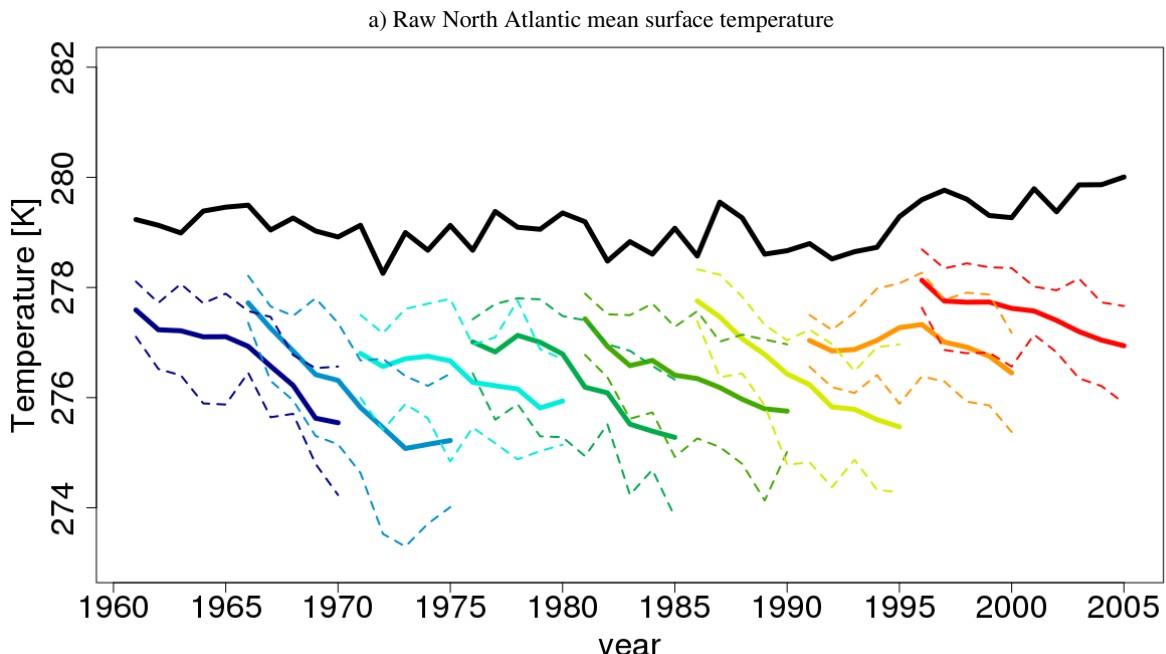

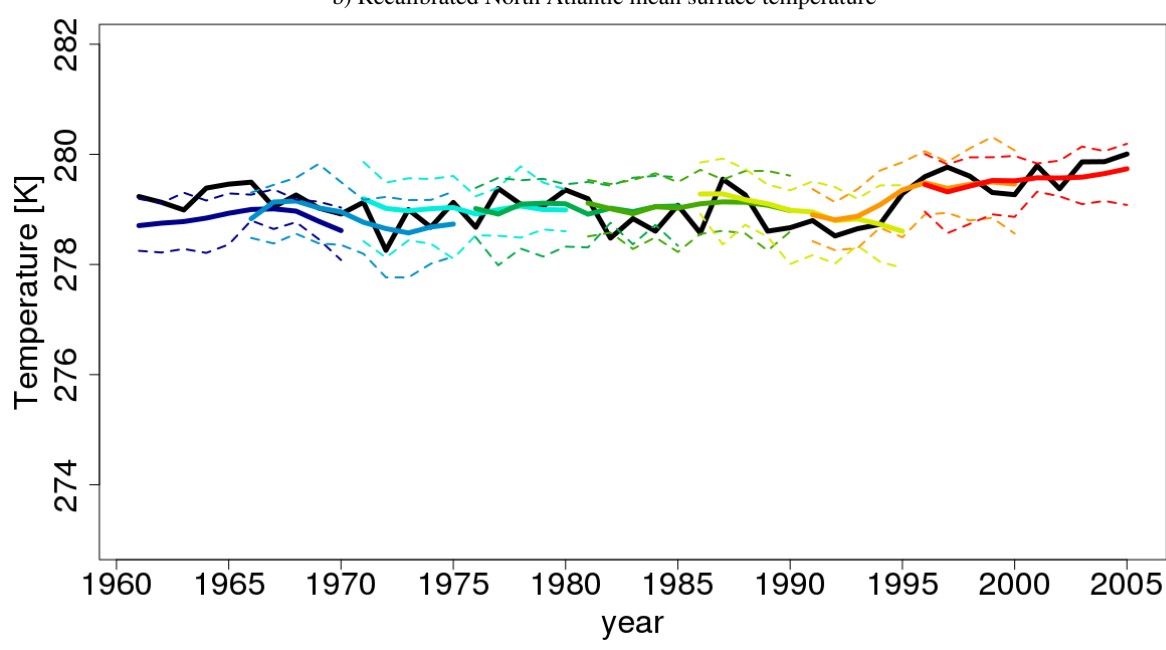

**Figure 5.** Temporal evolution of North Atlantic yearly mean surface temperature from MiKlip Prototype before (a) and (b) after recalibration with *DeFoReSt*. Shown are different start years with 5 year intervals (colored lines). The black line represents the surface temperature anomalies of NCEP 20CR. Units are in Kelvin [K].



**Figure 6.** Reliability (a), MSE (b), Ensemble Variance (c), and CRPSS (d) of surface temperature over the North Atlantic without any correction (black line), after drift correction (red line) and recalibration with *DeFoReSt* (blue line). The CRPSS for the raw forecasts (black line) is smaller than -1 and therefore not shown. As the drift correction method does not account for the ensemble spread, the ensemble variance of the raw model and the drift corrected forecast are equal. The vertical bars show the 95% confidence interval due 1000-wise bootstrapping.




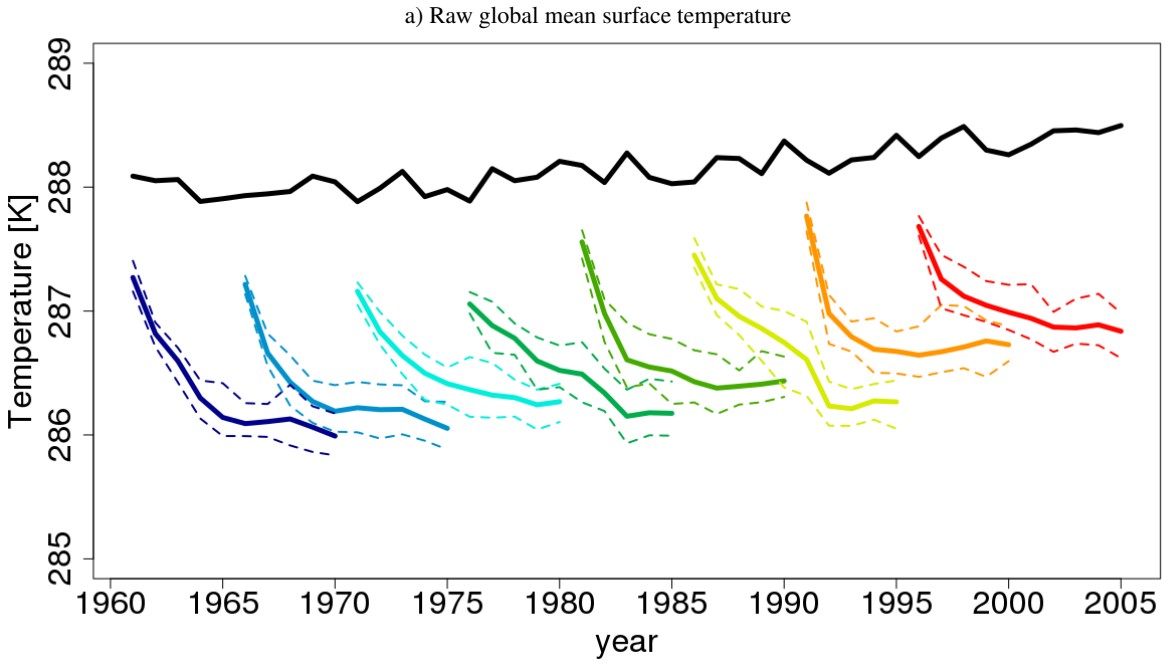

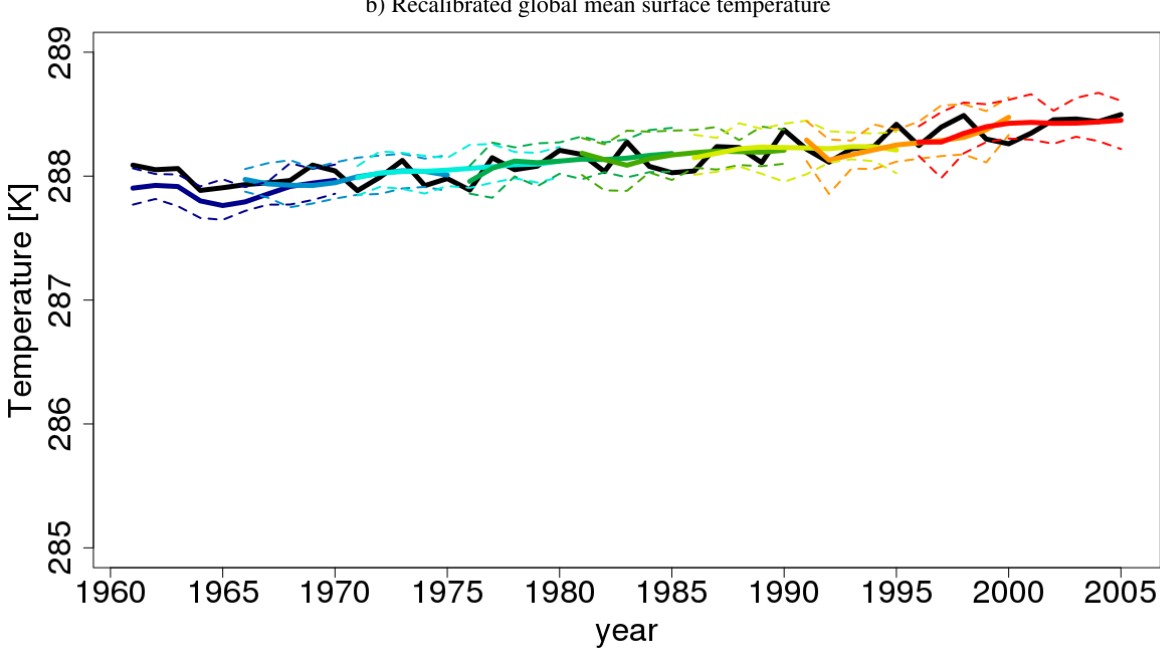

**Figure 7.** Temporal evolution of global yearly mean surface temperature from MiKlip Prototype (a) before and (b) after recalibration with *DeFoReSt*. Shown are different start years with 5 year intervals (colored lines). The black line represents the surface temperature anomalies of NCEP 20CR. Units are in Kelvin [K].





**Figure 8.** Reliability (a), MSE (b), Ensemble Variance (c), and CRPSS (d) of global mean surface temperature without any correction (black line), after drift correction (red line) and recalibration with *DeFoReSt* (blue line). The CRPSS for the raw forecasts (black line) is smaller than -1 and therefore not shown. As the drift correction method does not account for the ensemble spread, the ensemble variance of the raw model and the drift corrected forecast are equal. The vertical bars show the 95% confidence interval due 1000-wise bootstrapping.



|       | l=0   | l=1    | l=2  | l=3      | l=4   | l=5     | l=6   | l=7       |
|-------|-------|--------|------|----------|-------|---------|-------|-----------|
| $a_l$ | -0.61 | 0.0025 | 0.29 | -0.00046 | -0.11 | 0.0011  | 0.021 | -0.00029  |
| $b_l$ | 0.13  | 0.006  | 0.23 | -0.0027  | -0.12 | 0.00097 | 0.025 | -0.000197 |
| $w_l$ | 0.3   | 0      | 0.1  | 0.0014   | 0.01  | 0.0001  | 0     | 0         |

**Table A1.** Overview of the values coefficients $a_l$, $b_l$ and $w_l$.



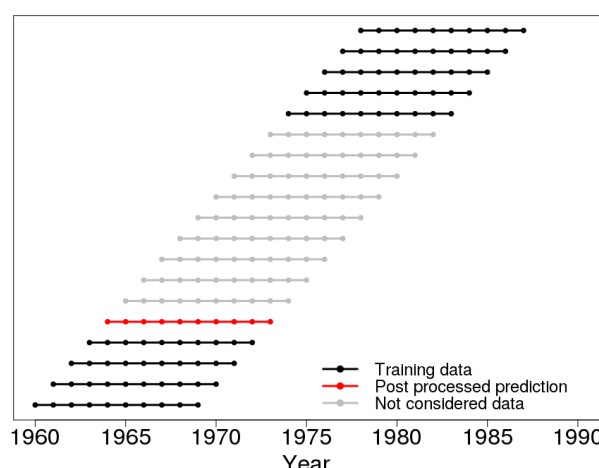

**Figure A1.** Schematic overview of the applied cross-validation procedure for a decadal climate prediction, initialized in 1964 (red dotted line). All hindcasts which are initialized outside the prediction period are used as training data (black dotted lines). A hindcast which is initialized inside the prediction period is not used for training (gray dotted lines).