# Peer review of "Parametric Decadal Climate Forecast Recalibration (DeFoReSt 1.0)"

_Geoscientific Model Development, 2017_

## Referee Comment (RC1) · Anonymous Referee #1 · 24 Aug 2017

**1   General comments**

This study proposes a probabilistic forecast recalibration scheme designed specifically to address the issues associated with decadal climate forecasts. Polynomial forms are estimated for the dependence of unconditional bias, conditional bias and ensemble spread on lead-time. The parameters of the polynomial forms are allowed to depend linearly on forecast-time, in order to account for any discrepancy between the long-term trends in the observations and the forecasts. The resulting recalibration is shown to outperform raw forecasts of surface temperature, and forecasts corrected only for lead-time dependent unconditional bias.

The extension of lead-time dependent bias correction to the conditional bias and en-

semble spread of parametric probability forecasts represents a valuable contribution to the decadal forecasting literature. Overall, the paper is clear and well written. The conclusions regarding the performance of the proposed recalibration are broadly justified by the results shown.

**2 Specific comments**

The review of existing recalibration methods is rather brief and would benefit from expansion, although it does cover the most relevant references. Other closely related methods for seasonal-to-decadal forecast recalibration include Eade et al. (2014, doi:10.1002/2014GL061146), Sansom et al. (2016, doi:10.1175/JCLI-D-15-0868.1) and references therein.

The DeFoReSt method addresses lead-time dependent unconditional and conditional biases in the ensemble mean, and unconditional bias in the ensemble spread. The authors cite the study by Fučkar et al. (2014) as a method specifically tailored to decadal forecasts. That study addressed corrections based on observed conditions at the time of forecast initialisation. Can the authors comment on the relevance of such corrections and why they chose to prioritise the biases addressed in DeFoReSt?

The orders of the polynomial forms used to capture lead-time dependence are fixed a priori. The authors acknowledge the need for a systematic method of selecting the order of the polynomial forms. However, it would be useful if they could provide some insight or justification for the choices they made?

A bootstrapping method is used to assess the uncertainty in the skill of the recalibrated forecasts. Can the authors clarify how the bootstrapping was performed?

In Section 5.2, the authors state that "After applying DeFoReSt, [...] the ensemble spread is fairly constant for all lead times". This statement is broadly supported by the

results in Figures 8 and 9, but runs contrary to the intuition that our uncertainty about the future climate should increase with lead-time. Can the authors comment on this surprising result?

**3  Technical corrections**

Several examples: choose Var(x) or var(x), there are examples of both
Several examples:  E[x] -> E(x) for consistency with var(x) and general bracket conventions
Several examples: e.g. -> e.g.,
Several examples: i.e. -> i.e.,
Page 12, Lines 4  15: yields to
Page 10, Line 26; Page 12, Line 6; Page 13, Line 23: yielding to

Page 2, Line 31: verification calibration -> verification and calibration
Page 3, Line 23: verification -> observations
Page 3, Line 26: calibration -> recalibration
Page 6, Line 11: the approach (9) -> the second term in (9)
Page 6, Line 13: It might be worth explaining in words that the dependence on lead-time is cubic, while the correction for errors in time trends is only linear.
Page 6, Line 19: Additionally, -> In addition
Page 6, Line 26: These assumption -> These assumptions
Page 6, Line 27: order selection will be topic of -> order selection will be a topic of
Page 7, Line 13: ensured -> guaranteed
Page 7, Line 21: this observations -> these observations
Page 8, Line 04: $\mu_x$ and $\epsilon_x$ -> The processes $\mu_x$ and $\epsilon_x$
Page 8, Line 13: Remove paragraph break

Page 8, Line 13: concrete -> specific or exact

Page 8, Line 13: this variability -> state which variability exactly

Page 8, Line 14: As for the recalibration strategy -> For recalibration,

Page 8, Line 15: we use. . . -> This sentence doesn't make sense.

Page 8, Line 21: In general, ensemble mean and ensemble variance both can dependent on lead time $\tau$ and start time t. -> In general, the ensemble mean and variance can both depend on lead time $\tau$ and start time t

Equation 21: It would be helpful to explain the motivation for non-linear form chosen here and the restriction in line 26.

Page 9, Lines 4-8: I understand from this paragraph that a trend is included in only the observations or the forecasts, but I am not clear on which. This needs to be clarified and possible linked explicitly to Eqn. 17 and Page 8, Lines 13-15.

Page 9, Line 26: 10 lead years $\tau$ = 1,...,10 -> 10 lead years ($\tau$ = 1,...,10)

Page 10, Line 24: which ESS values are lower one -> whose ESS values are less than one

Page 10, Line 25: The reduced performance. . . -> The lower performance

Page 11, Lines 25: model data with a low potential predictability

Page 12, Line 01: the worse MSE performance -> the lower MSE performance

Page 12, Line 03: On the contrary -> In contrast

Page 12, Line 08: bias within the -> bias in the

Page 12, Line 09: w.r.t. -> compared to

Page 12, Line 28: applied on surface -> applied to surface

Page 12, Line 27: global mean and a spatial mean

Page 13, Line 11: On the contrary, the raw model's -> The raw model's

Page 13, Line 26: is small than -> is less than

Page 13, Line 27: DeFoReSt slightly performs better -> DeFoReSt performs slightly better

Page 13, Line 32: with the corresponding -> and the corresponding

Page 14, Line 10: is constant -> is almost constant

[Figure]

Page 14, Line 13: perform equally -> perform similarly
Page 14, Line 20: also outperforming -> also outperform
Page 14, Line 21: climatology is solely not significant -> climatology is not significant
Page 15, Line 05: Analog to -> Following
Page 15, Line 13: impose -> imposed
Page 15, Line 14: conditional bias and ensemble spread dispersion-> conditional bias or ensemble spread

―――――――――――――――――

---

## Referee Comment (RC2) · Anonymous Referee #2 · 3 Oct 2017

General comments

This paper presents what may be a very important work in the study of decadal climate prediction. The authors presented the methodologically oriented post-processing model "Parametric Decadal Climate Forecast Recalibration (DeFoReSt)," to correct decadal climate prediction. The method uses earlier published approaches, and extend it to the long-term prediction by allowing the forecast errors to depend on forecast lead time. The performance of the presented approach is well established using different assessment measures.

[Figure]

Specific comments

1. It is mentioned in section 3.3 on lines 1-2 that the parameters are estimated by minimizing the average CRPS over the training period. Does this mean only the portion (the training) of the data was used for estimating the parameters? if yes, why the whole data was not used? it is expected that a training-validation grouping of data for checking the performance of DeFoReSt. But a parameter within DeFoReSt has to be primarily estimated from the whole data.

2. DeFoReSt was defined based on ensemble mean and variance functions (by my reading on the paper), where 15 ensemble members were selected. However, systematic errors vary widely between ensemble members, a simple ensemble averaging limits the relevance of DeFoReSt for long-term prediction. Arisido et al (2017) and Tebaldi et al (2005), see below, demonstrated that the common ensemble averaging method where each ensemble member has the same weight poses serious issues. I encourage the authors to discuss this issue supporting their argument with these papers and take into account the advisories in their revision.
- Arisido, M.W., Gaetan, C., Zanchettin, D. et al. Stoch Environ Res Risk Assess (2017). https://doi.org/10.1007/s00477-017-1383-2

- Tebaldi C, Smith RL, Nychka D, Mearns LO (2005) Quantifying uncertainty in projections of regional climate change: A Bayesian approach to the analysis of multimodel ensembles. Journal of Climate 18:1524-1540

3. How are the orders chosen for the polynomials used in equations such as (9) and (10). Without some cross-validation study it is not clear how a third order polynomial suffices for the drift along lead time. A sensitivity analysis for different order scenarios should guide for closer to the optimal choice needed for capturing the underlying features in a data.

Technical corrections

1. line 19 page 2, the acronym "PDF" should be defined on the first use.

2. line 1 Page 15, "..to change polynomially.." Remove "polynomially", then it is clear that the conditional bias and the ensemble dispersion change with lead time, while they change linearly with start time.

3. $\upsilon_5$ in (A9) page 17, the coefficient of $\tau^2$, is typos error?
* * *

---

## Author Comment (AC1) · 13 Oct 2017

**Answer to referee 1**

Thank you very much for your informative and detailed comments.

**General comments**

"This study proposes a probabilistic forecast recalibration scheme designed specifically to address the issues associated with decadal climate forecasts. Polynomial forms are

estimated for the dependence of unconditional bias, conditional bias and ensemble spread on lead-time. The parameters of the polynomial forms are allowed to depend linearly on forecast-time, in order to account for any discrepancy between the long-term trends in the observations and the forecasts. The resulting recalibration is shown to outperform raw forecasts of surface temperature, and forecasts corrected only for lead-time dependent unconditional bias. The extension of lead-time dependent bias correction to the conditional bias and ensemble spread of parametric probability forecasts represents a valuable contribution to the decadal forecasting literature. Overall, the paper is clear and well written. The conclusions regarding the performance of the proposed recalibration are broadly justified by the results shown."

**Specific comments**

1. "The review of existing recalibration methods is rather brief and would benefit from expansion, although it does cover the most relevant references. Other closely related methods for seasonal-to-decadal forecast recalibration include Eade et al. (2014, doi:10.1002/2014GL061146), Sansom et al. (2016, doi:10.1175/JCLI-D-15-0868.1) and references therein."

   **Answer:** Thank you for that advice. An extended review of existing (decadal) recalibration approaches will be added to the manuscript.

2. "The DeFoReSt method addresses lead-time dependent unconditional and conditional biases in the ensemble mean, and unconditional bias in the ensemble spread. The authors cite the study by Fučkar et al. (2014) as a method specifically tailored to decadal forecasts. That study addressed corrections based on observed conditions at the time of forecast initialisation. Can the authors comment on the relevance of such corrections and why they chose to prioritise the biases addressed in DeFoReSt?"

**Answer:** The approach of Fučkar et al. (2014) is highly relevant as the model bias and forecast skill depend on the initial conditions. This approach seems to outperform the bias correction approach based on a linear start year dependency. But the success of this approach may be dependent on the analyzed variable and region. Thus, in this first description of DeFoReSt we have decided for the less sophisticated approach proposed by Kharin et al. (2012) and Krusche et al. (2015) which assumes a linear start year dependency. However, combining the approach of Fučkar et al. (2014) with our approach is definitely an interesting idea for further studies.

-Fučkar, N. S., Volpi, D., Guemas, V., and Doblas-Reyes, F. J.: A posteriori adjustment of near-term climate predictions: Accounting for the drift dependence on the initial conditions, Geophysical Research Letters, 41, 5200–5207, 2014.

-Kharin, V. V., Boer, G. J., Merryfield, W. J., Scinocca, J. F., and Lee, W.-S.: Statistical adjustment of decadal predictions in a changing climate, Geophysical Research Letters, 39, 2012.

-Kruschke, T., Rust, H. W., Kadow, C., Müller, W. A., Pohlmann, H., Leckebusch, G. C., and Ulbrich, U.: Probabilistic evaluation of decadal prediction skill regarding Northern Hemisphere winter storms, Meteor. Z, 01, –, https://doi.org/10.1127/metz/2015/0641, 2015

3. "The orders of the polynomial forms used to capture lead-time dependence are fixed a priori. The authors acknowledge the need for a systematic method of selecting the order of the polynomial forms. However, it would be useful if they could provide some insight or justification for the choices they made?"

   **Answer:** For the proposed first version of DeFoReSt we follow the suggestion of Gangstø et al. (2013) and use a third order polynomial addressing the unconditional and conditional bias and a second order addressing the ensemble dispersion. With an increasing order of the polynomial the flexibility of the fitted

curve increases, while the parameter uncertainty also increases. Here, Gangstø et al. (2013) suggested that a third order polynomial is a good compromise between flexibility and parameter uncertainty. For the correction of the ensemble dispersion we assumed that a higher flexibility may not be necessary, because the MSE – which influences the dispersion – is already addressed by a third order polynomial of unconditional and conditional bias. Without any systematic model selection approach (which is to be developed) an answer to this question cannot be made more objective.

-Gangstø, R., A.P. Weigel, M.A. Liniger, C. Appenzeller, 2013: Methodological aspects of the validation of decadal predictions. – Climate Res. 55, 181–200, DOI: 10.3354/cr01135.

4. "A bootstrapping method is used to assess the uncertainty in the skill of the recalibrated forecasts. Can the authors clarify how the bootstrapping was performed?"

   **Answer:** The scores have been calculated for a period from 1961 to 2005. For bootstrapping we draw a new pair of dummy time series with replacement from the original validation period and calculate these scores again. Here, this procedure has been repeated 1000 times. An explanation will be added in the manuscript.

5. "In Section 5.2, the authors state that "After applying DeFoReSt, [...] the ensemble spread is fairly constant for all lead times". This statement is broadly supported by the results in Figures 8 and 9, but runs contrary to the intuition that our uncertainty about the future climate should increase with lead-time. Can the authors comment on this surprising result?"

   **Answer:** Maybe, the declaration "fairly constant" is misleading and s.th. like "basically constant" is more appropriate. In Fig. 5b and 7b the ensemble spread appears to be constant for all lead years, but Fig. 6c and 8c show that the ensemble spread of the recalibrated forecast increases from lead year 1 to lead

year 10. We admit that the ensemble spread of lead years 3-7 or 3-5 show an unexpected behavior by decreasing with lead years. Here, the ESS shows that the ensemble spread of the recalibrated forecast should be higher.

**Technical corrections**

1. Several examples: choose Var(x) or var(x), there are examples of both
   **Answer:** Will be corrected

2. Several examples: E[x] → E(x) for consistency with var(x) and general bracket conventions
   **Answer:** Will be corrected

3. Several examples: e.g. → e.g.,
   **Answer:** Will be corrected

4. Several examples: i.e. → i.e.,
   **Answer:** Will be corrected

5. Page 12, Lines 4 15: yields to
   **Answer:** Will be corrected

6. Page 10, Line 26; Page 12, Line 6; Page 13, Line 23: yielding to
   **Answer:** Will be corrected

7. Page 2, Line 31: verification calibration → verification and calibration
   **Answer:** Will be corrected

8. Page 3, Line 23: verification → observations

   **Answer:** Will be corrected

9. Page 3, Line 26: calibration → recalibration

   **Answer:** Will be corrected

10. Page 6, Line 11: the approach (9) → the second term in (9)

    **Answer:** Will be corrected

11. Page 6, Line 13: It might be worth explaining in words that the dependence on lead-time is cubic, while the correction for errors in time trends is only linear.

    **Answer:** A short explanation will be given

12. Page 6, Line 19: Additionally, → In addition

    **Answer:** Will be corrected

13. Page 6, Line 26: These assumption → These assumptions

    **Answer:** Will be corrected

14. Page 6, Line 27: order selection will be topic of → order selection will be a topic of

    **Answer:** Will be corrected

15. Page 7, Line 13: ensured → guaranteed

    **Answer:** Will be corrected

16. Page 7, Line 21: this observations → these observations

    **Answer:** Will be corrected

17. Page 8, Line 04: $\mu_x$ and $\varepsilon_x$ → The processes $\mu_x$ and $\varepsilon_x$

    **Answer:** Will be corrected

18. Page 8, Line 13: concrete → specific or exact

    **Answer:** Will be corrected

19. Page 8, Line 13: this variability → state which variability exactly

    **Answer:** Will be corrected

20. Page 8, Line 14: As for the recalibration strategy → For recalibration,

    **Answer:** Will be corrected

21. Page 8, Line 15: we use... → This sentence doesn't make sense.

    **Answer:** We will revise that sentence in the manuscript.

22. Page 8, Line 21: In general, ensemble mean and ensemble variance both can dependent on lead time $\tau$ and start time t. → In general, the ensemble mean and variance can both depend on lead time $\tau$ and start time t.

    **Answer:** Will be corrected

23. Equation 21: It would be helpful to explain the motivation for non-linear form chosen here and the restriction in line 26.

    **Answer:** The motivation is that both conditional bias and $\mu_x$ generally can depend on lead time and start time. The restriction is necessary to avoid negative values of the ensemble variance in Eq. 22. An explanation will be given in the manuscript.

24. Page 9, Lines 4-8: I understand from this paragraph that a trend is included in only the observations or the forecasts, but I am not clear on which. This needs to be clarified and possible linked explicitly to Eqn. 17 and Page 8, Lines 13-15.

**Answer:** A linear trend will be imposed on the pseudo-forecast due to the unconditional bias. Indeed, this needs to be clarified in the manuscript.

25. Page 9, Line 26: 10 lead years $\tau$ 1,...,10 → 10 lead years ($\tau$ = 1,...,10)

    **Answer:** Will be corrected

26. Page 10, Line 24: which ESS values are lower one → whose ESS values are less than one

    **Answer:** Will be corrected

27. Page 10, Line 25: The reduced performance...→ The lower performance

    **Answer:** Will be corrected

28. Page 11, Lines 25: model data with a low potential predictability

    **Answer:** Will be corrected

29. Page 12, Line 01: the worse MSE performance → the lower MSE performance

    **Answer:** Will be corrected

30. Page 12, Line 03: On the contrary → In contrast

    **Answer:** Will be corrected

31. Page 12, Line 08: bias within the → bias in the

    **Answer:** Will be corrected

32. Page 12, Line 09: w.r.t. → compared to

    **Answer:** Will be corrected

33. Page 12, Line 28: applied on surface → applied to surface

    **Answer:** Will be corrected

34. Page 12, Line 27: global mean and a spatial mean

    **Answer:** Will be corrected

35. Page 13, Line 11: On the contrary, the raw model's → The raw model's

    **Answer:** Will be corrected

36. Page 13, Line 26: is small than → is less than

    **Answer:** Will be corrected

37. Page 13, Line 27: DeFoReSt slightly performs better → DeFoReSt performs slightly better

    **Answer:** Will be corrected

38. Page 13, Line 32: with the corresponding → and the corresponding

    **Answer:** Will be corrected

39. Page 14, Line 10: is constant → is almost constant

    **Answer:** Will be corrected

40. Page 14, Line 13: perform equally → perform similarly

    **Answer:** Will be corrected

41. Page 14, Line 20: also outperforming → also outperform

    **Answer:** Will be corrected

42. Page 14, Line 21: climatology is solely not significant → climatology is not significant

    **Answer:** Will be corrected

43. Page 15, Line 05: Analog to → Following

    **Answer:** Will be corrected

44. Page 15, Line 13: impose → imposed

    **Answer:** Will be corrected

45. Page 15, Line 14: conditional bias and ensemble spread dispersion→ conditional bias or ensemble spread

    **Answer:** Will be corrected

---

## Author Comment (AC2) · 13 Oct 2017

**Answer to referee 2**

Thank you very much for your informative and detailed comments.

**General comments**

"This paper presents what may be a very important work in the study of decadal climate prediction. The authors presented the methodologically oriented post-processing

model "Parametric Decadal Climate Forecast Recalibration (DeFoReSt)," to correct decadal climate prediction. The method uses earlier published approaches, and extend it to the long-term prediction by allowing the forecast errors to depend on forecast lead time. The performance of the presented approach is well established using different assessment measures."

**Specific comments**

1. "It is mentioned in section 3.3 on lines 1-2 that the parameters are estimated by minimizing the average CRPS over the training period. Does this mean only the portion (the training) of the data was used for estimating the parameters? if yes, why the whole data was not used? it is expected that a training-validation grouping of data for checking the performance of DeFoReSt. But a parameter within DeFoReSt has to be primarily estimated from the whole data."

   **Answer:** In case of a validation/comparison with a reference data set (e.g., climatology or raw model) the training data set is only a portion of the whole available data set, while the remaining data is used for validation. We aim at estimating a forecast error for a setting comparable to the operational forecast situation where no observations for the forecast period is available. DeFoReSt parameters can only be estimated from the available observataion period but the re-calibration is carried out on the forecast period, i.e. outside the period used for parameter estimation. Hence using the full hindcast period for estimating parameters and obtain a "forecast error" for hindcast from the same period would lead to overestimation of skill. Parameter estimation using the full available data set could be used once we use DeFoReSt for re-calibration decadal forecasts, for e.g., 2018-2027.

2. "DeFoReSt was defined based on ensemble mean and variance functions (by my reading on the paper), where 15 ensemble members were selected. However,

sys- tematic errors vary widely between ensemble members, a simple ensemble averaging limits the relevance of DeFoReSt for long-term prediction. Arisido et al (2017) and Tebaldi et al (2005), see below, demonstrated that the common ensemble averaging method where each ensemble member has the same weight poses serious issues. I encourage the authors to discuss this issue supporting their argument with these pa- pers and take into account the advisories in their revision.

- Arisido, M.W., Gaetan, C., Zanchettin, D. et al. Stoch Environ Res Risk Assess (2017). https://doi.org/10.1007/s00477-017-1383-2 - Tebaldi C, Smith RL, Nychka D, Mearns LO (2005) Quantifying uncertainty in projec- tions of regional climate change: A Bayesian approach to the analysis of multimodel ensembles. Journal of Climate 18:1524-1540"

**Answer:** It is true that the ensemble members of a multi-model ensemble cannot be treated equally because every corresponding model has different systematic errors. However, in this study we apply DeFoReSt to a single model ensemble with 15 members generated by lagged-day-initialization from MPI-ESM-LR; i.e., we do not expect that the single ensemble members have different systematic errors (due to the model). Nonetheless, for a recalibration of a multimodel ensemble DeFoReSt needs to be adapted. Which would be a topic of further research.

3. "How are the orders chosen for the polynomials used in equations such as (9) and (10). Without some cross-validation study it is not clear how a third order polynomial suffices for the drift along lead time. A sensitivity analysis for different order scenar- ios should guide for closer to the optimal choice needed for capturing the underlying features in a data."

**Answer:** We agree that there is need for a transparent model selection strategy! As already mentioned in section 6, this will be topic for future studies. For the first version of DeFoReSt we follow the suggestion of Gangstø et al. (2013) and use

a third order polynomial addressing the unconditional and conditional bias and a second order addressing the ensemble dispersion. With an increasing order of the polynomial the flexibility of the fitted curve increases, while the parameter uncertainty also increases. Here, Gangstø et al. (2013) suggested that a third order polynomial is a good compromise between flexibility and parameter uncertainty. For the correction of the ensemble dispersion we assumed that a higher flexibility may not be necessary, because the MSE -which influences the dispersion- is addressed by a third order polynomial of unconditional and conditional bias.

-Gangstø, R., A.P. Weigel, M.A. Liniger, C. Appenzeller, 2013: Methodological aspects of the validation of decadal predictions. – Climate Res. 55, 181–200, DOI: 10.3354/cr01135.

**Technical corrections**

1. "line 19 page 2, the acronym "PDF" should be defined on the first use"

   **Answer:** Will be corrected

2. "line 1 Page 15, "..to change polynomially.." Remove "polynomially", then it is clear that the conditional bias and the ensemble dispersion change with lead time, while they change linearly with start time."

   **Answer:** Will be corrected

3. "$v_5$ in (A9) page 17, the coefficient of $\tau^2$, is typos error?"

   **Answer:** Indeed, it should be $b_5$ instead of $v_5$.